# IMPROVING TRANSFORMER INTERPRETABILITY WITH ACTIVATION CONTRAST-BASED ATTRIBUTION

## ABSTRACT

Transformers have revolutionized AI research, particularly in natural language processing (NLP). However, understanding the decisions made by transformer-based models remains challenging, which impedes trust and safe deployment in real-world applications. While activation-based attribution methods have proven effective in explaining transformer-based text classification models, our findings suggest that they may suffer from class-irrelevant features within activations, potentially degrading the quality of their interpretations. To address this issue, we introduce Contrast-CAT, a novel activation contrast-based attribution method that improves token-level attribution by filtering out class-irrelevant features from activations. Contrast-CAT enhances interpretability by contrasting the activations of input sequences with reference activations, allowing for the generation of clearer and more faithful attribution maps. Our experiments demonstrate that Contrast-CAT consistently outperforms state-of-the-art methods across various datasets and models, achieving significant gains over the second-best methods with average improvements in AOPC and LOdds by $\times 1.30$ and $\times 2.25$, respectively, under the MoRF setting. Contrast-CAT provides a promising step forward in enhancing the interpretability and transparency of transformer-based models.

## 1 INTRODUCTION

The success of transformers (Vaswani et al., 2017), particularly in natural language processing (NLP), has been remarkable in recent years. This success has transcended both academic and industrial boundaries, integrating them more into our daily lives. Unfortunately, this integration has also increased the risk of direct exposure to AI errors, heightening the need to ensure the safety, security, and trustworthiness of AI models by promoting transparency in AI systems (The White House, 2023; Dunietz et al., 2024; European Commission, 2024). As a result, developing methods to interpret the decision-making processes of transformer-based models has become essential.

To meet this need, numerous methods have been proposed for interpreting transformer-based models, particularly for text classification, where they have demonstrated remarkable performance. These methods often provide attribution maps telling the relative contributions of input tokens to the model's decisions; in Section 2, we categorize them into attention-based, LRP-based, and activation-based attribution methods. This work focuses on activation-based attribution, which leverages a model's activation information to generate attribution maps, achieving state-of-the-art performance in attribution quality thus far.

In essence, activation-based attribution maps are created using activations from a certain layer or multiple layers of a neural network corresponding to an input sequence. Then, the output gradient of the prospective class with respect to the activations is imposed on the activations to extract only class-relevant features (Selvaraju et al., 2017; Qiang et al., 2022).

However, we found that this procedure can still be affected by class-irrelevant features present in activations, hindering the creation of accurate class-specific interpretations. For example, Figure 1 shows attribution maps generated by AttCAT in panel (A), one of the state-of-the-art activation-based attribution method (Qiang et al., 2022), for a movie review 'It is very slow.' classified as negative. We expect the word 'slow' to be detected as relevant, with a positive attribution value for the negative review. However, AttCAT fails to detect the word, being confused by the punctuation mark. To the contrary, our proposed method Contrast-CAT puts the highest attribution on 'slow'.

| (A) AttCAT | | | | | | (B) Contrast-CAT | | | | |
|---|---|---|---|---|---|---|---|---|---|---|
| Layer1 | 0.04 | -0.10 | 0.29 | -0.07 | -0.15 | Layer1 | -0.05 | -0.09 | 0.10 | 0.14 | -0.07 |
| Layer2 | -0.04 | -0.04 | -0.04 | -0.13 | 0.25 | Layer2 | -0.05 | -0.06 | -0.01 | 0.09 | -0.02 |
| Layer3 | 0.01 | -0.15 | 0.01 | 0.00 | 0.14 | Layer3 | 0.02 | -0.07 | 0.02 | 0.04 | -0.14 |
| Layer4 | 0.08 | -0.08 | -0.06 | -0.05 | 0.04 | Layer4 | 0.06 | -0.01 | -0.01 | 0.07 | -0.16 |
| Layer5 | 0.06 | 0.01 | -0.01 | -0.16 | 0.06 | Layer5 | 0.05 | 0.03 | 0.02 | 0.00 | -0.16 |
| Layer6 | 0.01 | 0.01 | 0.00 | -0.00 | 0.10 | Layer6 | -0.00 | -0.03 | -0.01 | 0.05 | 0.02 |
| Layer7 | 0.04 | -0.05 | -0.02 | -0.08 | 0.07 | Layer7 | 0.01 | -0.05 | -0.01 | 0.03 | 0.07 |
| Layer8 | 0.02 | 0.02 | 0.00 | -0.07 | -0.05 | Layer8 | -0.00 | -0.01 | 0.02 | -0.02 | -0.04 |
| Layer9 | 0.01 | -0.04 | -0.02 | 0.07 | -0.01 | Layer9 | 0.06 | 0.11 | 0.07 | 0.11 | 0.09 |
| Layer10 | 0.03 | 0.01 | -0.00 | -0.00 | -0.00 | Layer10 | 0.09 | 0.08 | 0.04 | 0.06 | 0.07 |
| Layer11 | 0.03 | 0.04 | 0.00 | 0.00 | -0.06 | Layer11 | 0.06 | 0.07 | 0.02 | 0.04 | 0.01 |
| Layer12 | 0.03 | 0.03 | 0.01 | 0.02 | -0.02 | Layer12 | 0.05 | 0.05 | 0.03 | 0.03 | 0.01 |
| Map | 0.33 | -0.33 | 0.17 | -0.47 | 0.36 | Map | 0.29 | 0.03 | 0.27 | 0.64 | -0.32 |
| | **It** | **is** | **very** | **slow** | **.** | | **It** | **is** | **very** | **slow** | **.** |

Figure 1: The heatmaps display attribution values from different encoder layers of the BERT$_{base}$ model and their corresponding attribution maps for a negative review prediction. These maps are generated by AttCAT (panel A), which applies gradients to activations, and Contrast-CAT (panel B), which applies gradients to activation contrast information. Values closer to 1 (red) indicate a stronger contribution to the negative prediction, while values closer to 0 indicate a weaker contribution.

In this paper, we propose Contrast-CAT, a novel activation-based attribution method for transformer-based text classification models. Contrast-CAT is designed to produce high-quality token-level attribution maps by filtering out class-irrelevant features from activations through our new activation-contrasting framework. Our experiments demonstrate that Contrast-CAT significantly improves the quality of token-level attribution.

**Contributions** Our contributions can be summarized as follows. (1) We observe that activation-based attribution methods for transformer-based text classification models may incorporate class-irrelevant features within activations, potentially degrading attribution quality. (2) We propose Contrast-CAT for generating token-level attribution maps based on a novel activation-contrasting framework tailored for transformer architecture. Unlike existing activation-based attribution methods, Contrast-CAT leverages differences between target and multiple reference activations to reduce class-irrelevant features in the target activation, thereby improving attribution quality. (3) We provide experimental results demonstrating that Contrast-CAT significantly outperforms state-of-the-art methods, achieving average improvements of ×1.30 and ×2.25 in AOPC and LOdds under the MoRF setting, and ×1.34 and ×1.03 under the LeRF setting, compared to the second-best methods.

## 2 RELATED WORK

We describe attribution methods for interpreting transformer-based text classification models, categorizing them into attention-based, LRP-based, and activation-based approaches.

**Attention-based Attribution** Attention-based attribution methods rely on attention scores, a key component of transformers (Vaswani et al., 2017). Under the assumption that input tokens with high attention scores significantly influence model outputs, numerous studies (Martins & Astudillo, 2016; Mullenbach et al., 2018; Clark et al., 2019; Abnar & Zuidema, 2020; Modarressi et al., 2022; Mohebbi et al., 2023) have employed attention scores for interpretative purposes of a model. Specifically, (Abnar & Zuidema, 2020) proposed Rollout, which integrates attention scores across multiple layers while accounting for skip connections in transformer architectures to capture information flow. Additionally, there have been many papers (Chrysostomou & Aletras, 2021; Barkan et al., 2021) that introduce the gradient of attention weight for interpretation. Despite advances in attention-based methods, significant debate remains about whether attention scores truly reflect the relevance of model predictions, as highlighted in (Jain & Wallace, 2019; Wiegreffe & Pinter, 2019).

**LRP-based Attribution** Layer-wise relevance propagation (LRP) (Bach et al., 2015) is a technique for backpropagating relevance scores through a neural network, with the scores reflecting our specific interest in the model's prediction. Building on LRP, several studies have derived explanations for model behavior (Gu et al., 2018; Voita et al., 2019; Chefer et al., 2021). In (Voita et al., 2019), LRP was partially used to determine the most important attention heads within a spe-

cific transformer's encoder layer, utilizing relevance scores for the attention weights. (Chefer et al., 2021) introduces TransAtt, which propagates relevance scores through all layers of a transformer, combining these scores with gradients of the attention weights and utilizing the Rollout technique for multi-layer integration. However, LRP-based methods are limited by certain assumptions, known as the LRP rules, designed to uphold the principle of relevance conservation (Montavon et al., 2019).

**Activation-based Attribution**  In contrast to the methods discussed above, activation-based attribution primarily relies on activation information from each layer of a transformer model. These methods are based on core ideas originally developed for convolutional neural networks (CNNs), which have been shown to be effective for generating high-quality interpretations with simple implementations and broad versatility (Selvaraju et al., 2017; Wang et al., 2020; Lee & Han, 2022). In (Qiang et al., 2022), the authors introduced AttCAT as the first adaptation of Grad-CAM (Selvaraju et al., 2017), one of the most popular activation-based methods for CNNs, to interpret the decisions of transformer-based text classification models. AttCAT generates token-level attribution maps by merging activations and their gradients in relation to the model's predictions, following Grad-CAM's essential approach, which uses gradients to reflect class-relevant information. Similarly, (Englebert et al., 2023) introduced TIS adapting Score-CAM (Wang et al., 2020): TIS uses the centroids of activation clusters identified from the activation from all layers to compute relevance scores in a manner akin to Score-CAM.

Although there are existing attribution methods for transformer-based text classification models that use gradients to extract class-relevant features from activations, no approach has yet focused on filtering out class-irrelevant features through activation contrasting to improve attribution quality.

## 3 PRELIMINARY

We discuss our problem setup and provide a brief overview of the transformer structure.

**Problem Statement**  Consider a pre-trained transformer-based model as a function $f$ processing input tokens $x := \{x_i\}_{i=1}^T$, where $T$ is the length of the input sequence, and each token is denoted as $x_i \in \mathbb{R}^n$. Our objective is to generate a token-level attribution map $I(x) := \{I(x)_i\}_{i=1}^T$, where $I(x)_i$ represents the relevance score of each input token $x_i$ regarding the output $f(x)$.

**Transformers**  Let us consider a transformer-based model which is composed of $L$ stacked layers of identical structure. We denote that the $\ell$-th layer outputs an activation sequence $A^\ell := \{A_i^\ell\}_{i=1}^T$ that corresponds to input tokens, where $A_i^\ell \in \mathbb{R}^n$. Each layer computes its output by combining the output from the attention layer with the previous layer's activation, where the attention layer calculates the attention scores:

$$\alpha^{\ell,h} := \text{softmax}\left(Q^{\ell,h}(A^{\ell-1}) \cdot K^{\ell,h}(A^{\ell-1})^T / \sqrt{d}\right). \tag{1}$$

Here, $Q^{\ell,h}(\cdot)$, $K^{\ell,h}(\cdot)$, and $V^{\ell,h}(\cdot)$ are the transformations for computing the query, key, and value of the $\ell$-th layer's $h$-th head, respectively, and $d$ is a scaling factor. $\alpha^{\ell,h} \in \mathbb{R}^{T \times T}$ refers to the attention map of the $h$-th head, which contains attention scores, where $h = 1 \ldots H$. We denote by $A^{\ell,h}$ the output of the $h$-th attention head in the $\ell$-th layer:

$$A^{\ell,h} := \alpha^{\ell,h} \cdot V^{\ell,h}(A^{\ell-1}).$$

The outputs from multiple attention heads are concatenated and then combined using a fully connected layer with the skip connection: $\hat{A}^\ell := \text{Concat}(A^{\ell,1}, A^{\ell,2}, \ldots, A^{\ell,H}) \cdot \tilde{W}^\ell + A^{\ell-1}$, where $\tilde{W}^\ell$ is the weight of the fully connected layer. Finally, the $\ell$-th layer's output $A^\ell \in \mathbb{R}^{T \times n}$ is computed using a feed-forward layer and skip connection:

$$A^\ell = \hat{A}^\ell \cdot W^\ell + \hat{A}^\ell, \tag{2}$$

where $W^\ell \in \mathbb{R}^{n \times n}$ is the weight for the feed-forward layer. We have omitted bias parameters and layer normalization in the above expressions for simplicity.

$$I_R(x) \begin{cases} I_R(x)_1 = \hat{\alpha}_1^1 \left[ \frac{\partial f_c(x)}{\partial A_1^1} \odot \underbrace{(A_1^1 - R_1^1)}_{\text{Contrasting}} \right] + \cdots + \hat{\alpha}_1^L \left[ \frac{\partial f_c(x)}{\partial A_1^L} \odot \underbrace{(A_1^L - R_1^L)}_{\text{Contrasting}} \right] \\ \vdots \qquad\qquad\qquad\qquad\qquad \vdots \\ I_R(x)_6 = \hat{\alpha}_6^1 \left[ \frac{\partial f_c(x)}{\partial A_6^1} \odot \underbrace{(A_6^1 - R_6^1)}_{\text{Contrasting}} \right] + \cdots + \hat{\alpha}_6^L \left[ \frac{\partial f_c(x)}{\partial A_6^L} \odot \underbrace{(A_6^L - R_6^L)}_{\text{Contrasting}} \right] \end{cases}$$

Figure 2: The construction of an attribution map $I_R(x)$ for an input token sequence $x$ by Contrast-CAT, using a single reference activation, is illustrated alongside the transformer architecture. The colors represent internal model components used to construct an attribution map: red for gradient information, yellow for attention information, and blue for reference activation.

## 4 CONTRAST-CAT

We introduce Contrast-CAT (activation Contrast-based Class Activation Token), a new token-level input attribution method for transformer architecture based on activation contrasting. Figure 2 provides a simplified illustration of the attribution map construction process for Contrast-CAT.

### 4.1 CONSTRUCTION OF ATTRIBUTION MAP

Suppose that $c$ is the prospective class of a given input token sequence $x$, for which the output of a transformer-based model is denoted by $f_c(x)$. For the activation map $A^\ell$ at the $\ell$-th layer of a neural network, we can adapt the result in Lee & Han (2022) so that $f_c(x)$ is to be approximated with respect to $A^\ell(x)$ based on the first-order Taylor expansion as follows:

$$f_c(x) \approx \sum_{i,j} \left( \frac{\partial f_c(x)}{\partial A^\ell} \odot (A^\ell(x) - \tilde{A}^\ell(\tilde{x})) \right)_{i,j}, \tag{3}$$

where $\tilde{A}^\ell$ is the activation of an input $\tilde{x}$ which satisfies $f_c(\tilde{x}) \approx 0$, $\frac{\partial f_c(x)}{\partial A^\ell} \in \mathbb{R}^{T \times n}$ represents the gradient of $f_c(x)$ with respect to $A^\ell$, and $\odot$ is the element-wise multiplication. Here, $i = 1, \ldots, T$ and $j = 1, \ldots, n$ can be considered as the indices over tokens and the elements of activation in the case of transformers, respectively. Inspired by this, we define our attribution map $I_R(x)$ as follows:

$$I_R(x)_i := \sum_{\ell=1}^{L} \hat{\alpha}_i^\ell \sum_{j=1}^{n} \left( \frac{\partial f_c(x)}{\partial A_i^\ell} \odot (A_i^\ell - R_i^\ell) \right)_j. \tag{4}$$

Here, $\hat{\alpha}_i^\ell \in \mathbb{R}$ is the averaged attention score for $i$-th token at $\ell$-th layer, defined as $\hat{\alpha}_i^\ell := \frac{1}{HT} \sum_{h=1}^{H} \sum_{j=1}^{T} \alpha_{i,j}^{\ell,h}$ for $\alpha_{i,j}^{\ell,h}$ defined in Eq. (1), and $H$ is the number of attention heads. In Figure 2, $\frac{\partial f_c(x)}{\partial A_i^\ell}$, $\hat{\alpha}_i^\ell$, and $R_i^\ell$ are depicted in red, yellow, and blue color, respectively.

**Contrastive References** For $\tilde{A}^\ell$ in Eq. (3), we choose a sequence of activations $R^\ell := \{R_i^\ell\}_{i=1}^{T}$ for which the corresponding input token sequence $r := \{r_i\}_{i=1}^{T}$ satisfies $f_c(r) < \gamma$ for the target class $c$ and a pre-defined small number $\gamma > 0$ (we used $\gamma = 10^{-3}$ in our experiments). We call $r$ and $R^\ell$ as a reference token sequence and the reference activation of the $\ell$-th layer, respectively.

We consider the reference activation $R$ to be contrastive to the target activation $A$ since $f_c(A(x))$ is high while $f_c(R(r))$ is low by construction. Our attribution map (Eq. (4)) uses the subtraction $A^\ell - R^\ell$ for building the attribution map, where we expect that the subtraction would remove features

of classes other than the target class $c$ inherent in $A^\ell$ and thereby reveal the important features in $x$ more vividly in attribution maps.

**Extraction from Multiple Layers**    As discovered in previous studies (Jawahar et al., 2019; Turton et al., 2021; Pascual et al., 2021), the semantic information of given input token sequences processed by transformer-based models varies across different layers, ranging from phrase-level information to deeper semantic meanings. Therefore, unlike traditional activation-based attribution methods for CNNs, which only use activations extracted from a single, usually the penultimate (Selvaraju et al., 2017; Lee & Han, 2022), layer, we use the activations $A^\ell$ in Eq. (2) from multiple layers, where $\ell = 1, \dots L$, along with their layer-wise attention scores $\alpha^{\ell,h}$ in Eq. (1) to capture layer-specific meanings for each token across various layers. This design allows us to reflect dominantly attended token-level information of the target activations across multiple layers by combining $\hat{\alpha}_i^\ell$.

Finally, by incorporating the gradient information $\frac{\partial f_c(x)}{\partial A^\ell} \in \mathbb{R}^{T \times n}$ element-wise, which quantifies how changes in each element of the activations $A^\ell$ affect the model's prediction, we can highlight the specific contributions of the target activations.

### 4.2 USE OF MULTIPLE CONTRASTS

The activation subtraction in Eq. (4) is done with a single reference belonging to a certain class. However, it would be beneficial to contrast with multiple references of various classes, considering that the target activation $A^\ell$ may contain features of more than one non-target class. Furthermore, features within the target activation that persist after subtraction with various reference activations are more likely to represent class-relevant features unique to the target activation. For this purpose, we create a set of attribution maps $D$ by conducting the previous procedure in Section 4.1 with multiple reference activation, where $D := \{I_{R(r)}(x) : r \in \text{training set}, f_c(R(r)) < \gamma\}$.

These reference activations can be sampled and cached during training and used later to generate attribution maps – we call this the reference library. We used such a reference library with 30 pre-computed references per class.

**Refinement with Multiple Contrasts**    We refine Contrast-CAT using the set $D$. Our refinement process involves selectively filtering out maps from $D$ that likely contain incorrect attributes. For this purpose, we assess the attribution quality of each map in the set $D$ and exclude those that do not meet our established criteria based on the assessed scores.

To evaluate the attribution quality of each map, we utilize a deletion test (Petsiuk, 2018; Wang et al., 2020; Lee & Han, 2022). This approach is adapted here as a token-wise deletion test. For each map in $D$, we calculate the average probability drop score by sequentially removing the top-ranked tokens based on their attribution values and comparing the model's output before and after each modification. This measures the decrease in the model's predictive probability due to the removal of each token. This procedure is conducted on a token-by-token basis, where each token's removal is individually assessed to determine its impact on the model's output. The average probability drop score is then computed by taking the mean of these individual probability drops, thereby quantifying the average quality of the attribution map for each token.

Finally, we generate Contrast-CAT by averaging over all the attribution maps:

$$I(x) := \frac{1}{|M|} \sum_{I_R(x) \in M} I_R(x),$$

where $M := \{I_R(x) \in D : S(I_R(x)) \geq \rho\}$. Here, $S(I_R(x))$ represents the average probability drop score of each map $I_R(x)$. In our experiments, we set the value of $\rho$ as the mean plus standard deviation of these scores from the set of attribution maps.

## 5 EXPERIMENTS

In all our experiments, we used PyTorch v.1.9.1, Numpy v.1.17.4, and scikit-learn v.0.22.2 libraries on the Ubuntu 18.04.3 (64-bit) system. The hardware configuration included an Intel CPU (Xeon Silver 4214), 32GB of memory, and an NVIDIA GPU (GeForce RTX2080Ti) with CUDA v.10.2.

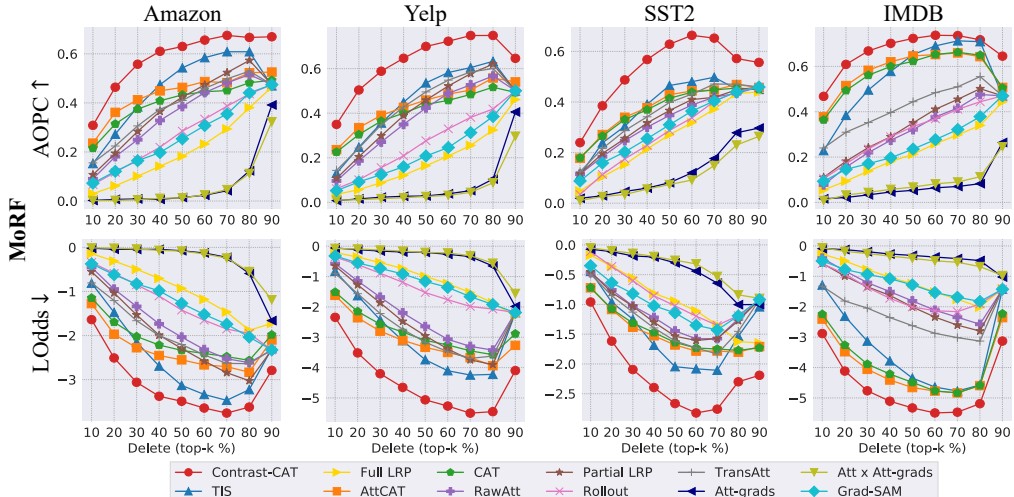

Figure 3: Quantitative comparison of the faithfulness evaluation of Contrast-CAT and other attribution methods, measured under the MoRF (Most Relevant First) setting. The arrows mean that ↑: higher is better, and ↓: lower is better.

**Experiment Settings**   We used the pre-trained BERT$_{base}$ model (Devlin et al., 2019), consisting of 12 encoder layers with 12 attention heads, as the transformer-based model for our experiments (see supplementary material for results using other transformer-based models). We used four popular datasets for text classification tasks: Amazon Polarity (Zhang et al., 2015), Yelp Polarity (Zhang et al., 2015), SST2 (Socher et al., 2013), and IMDB (Maas et al., 2011). We reported our results using 2000 random samples from the test sets of each dataset, except for SST2, for which the entire set was used since the entire dataset had fewer than 2000 samples.

We compared our method to various attribution methods, categorized by attention-based: RawAtt, Rollout (Abnar & Zuidema, 2020), Att-grads, Att×Att-grads, and Grad-SAM (Barkan et al., 2021); LRP-based: Full LRP (Ding et al., 2017), Partial LRP (Voita et al., 2019), and TransAtt (Chefer et al., 2021); and activation-based methods: CAT, AttCAT (Qiang et al., 2022), and TIS (Englebert et al., 2023). Open-source implementations from (Qiang et al., 2022) and (Englebert et al., 2023) were used for our experiments.

**Evaluation Metrics**   We used the area over the perturbation curve (denoted by AOPC) (Nguyen, 2018; Chen et al., 2020) and the log-odds (LOdds) (Shrikumar et al., 2017; Chen et al., 2020) metrics for assessing the faithfulness of attribution following the previous research (Qiang et al., 2022). Faithfulness refers to the accuracy with which an attribution map's scores reflect the actual influence of each input token on the model's prediction. The AOPC and LOdds metrics are defined as follows: (1) $\text{AOPC}(k) := \frac{1}{N} \sum_{i=1}^{N} (y_i^c - \tilde{y}_i^c)$, and (2) $\text{LOdds}(k) := \frac{1}{N} \sum_{i=1}^{N} \log \left( \frac{\tilde{y}_i^c}{y_i^c} \right)$. Here, $N$ is the total number of data points used for evaluation, and $y_i^c$ denotes the model's prediction probability for the class $c$ of a given input token sequence $x$, while $\tilde{y}_i^c$ indicates the probability after removing the top-$k\%$ of input tokens based on relevance scores from an attribution map.

To evaluate attribution quality more precisely using the AOPC and LOdds metrics, and to address inconsistencies in evaluation results caused by the order of token removal (i.e., removing the most relevant tokens first versus the least relevant tokens first) (Rong et al., 2022), we conducted experiments under two settings: one where tokens were removed in descending order of relevance scores (MoRF: Most Relevant First), and another in ascending order (LeRF: Least Relevant First). Consistently achieving high-quality attribution under both conditions indicates superior attribution quality. Specifically, under the MoRF setting, higher AOPC and lower LOdds indicate better attribution, while under the LeRF setting, lower AOPC and higher LOdds suggest better performance.

| (A) MoRF (Most Relevant First) | | | | | | | | |
|---|---|---|---|---|---|---|---|---|
| Dataset | Amazon | | Yelp | | SST2 | | IMDB | |
| Method | AOPC↑ | LOdds↓ | AOPC↑ | LOdds↓ | AOPC↑ | LOdds↓ | AOPC↑ | LOdds↓ |
| RawAtt | 0.424 | 0.405 | 0.412 | 0.462 | 0.386 | 0.471 | 0.335 | 0.564 |
| Rollout | 0.327 | 0.516 | 0.282 | 0.601 | 0.329 | 0.558 | 0.339 | 0.566 |
| Att-grads | 0.061 | 0.749 | 0.059 | 0.754 | 0.132 | 0.691 | 0.061 | 0.759 |
| Att×Att-grads | 0.054 | 0.756 | 0.045 | 0.763 | 0.109 | 0.711 | 0.075 | 0.746 |
| Grad-SAM | 0.312 | 0.526 | 0.235 | 0.633 | 0.356 | 0.518 | 0.266 | 0.623 |
| Full LRP | 0.242 | 0.592 | 0.190 | 0.652 | 0.310 | 0.538 | 0.233 | 0.631 |
| Partial LRP | 0.463 | 0.356 | 0.447 | 0.422 | 0.400 | 0.461 | 0.364 | 0.538 |
| TransAtt | 0.461 | 0.366 | 0.473 | 0.404 | 0.432 | 0.428 | 0.458 | 0.455 |
| CAT | 0.482 | 0.341 | 0.440 | 0.383 | 0.452 | 0.382 | 0.632 | 0.215 |
| AttCAT | 0.527 | 0.292 | 0.470 | 0.346 | 0.461 | 0.372 | 0.644 | 0.198 |
| TIS | 0.560 | 0.241 | 0.494 | 0.349 | 0.463 | 0.367 | 0.618 | 0.277 |
| Contrast-CAT | **0.703** | **0.117** | **0.687** | **0.131** | **0.654** | **0.157** | **0.738** | **0.101** |
| (B) LeRF (Least Relevant First) | | | | | | | | |
| Dataset | Amazon | | Yelp | | SST2 | | IMDB | |
| Method | AOPC↓ | LOdds↑ | AOPC↓ | LOdds↑ | AOPC↓ | LOdds↑ | AOPC↓ | LOdds↑ |
| RawAtt | 0.133 | 0.694 | 0.093 | 0.723 | 0.249 | 0.577 | 0.158 | 0.688 |
| Rollout | 0.166 | 0.670 | 0.130 | 0.687 | 0.373 | 0.448 | 0.126 | 0.711 |
| Att-grads | 0.636 | 0.186 | 0.560 | 0.252 | 0.601 | 0.223 | 0.588 | 0.271 |
| Att×Att-grads | 0.707 | 0.111 | 0.660 | 0.145 | 0.681 | 0.126 | 0.709 | 0.127 |
| Grad-SAM | 0.139 | 0.677 | 0.107 | 0.713 | 0.285 | 0.547 | 0.118 | 0.715 |
| Full LRP | 0.254 | 0.588 | 0.187 | 0.649 | 0.377 | 0.454 | 0.199 | 0.656 |
| Partial LRP | 0.122 | 0.700 | 0.088 | 0.725 | 0.237 | 0.585 | 0.134 | 0.701 |
| TransAtt | 0.089 | 0.731 | 0.063 | 0.751 | 0.215 | 0.605 | 0.061 | 0.761 |
| CAT | 0.108 | 0.712 | 0.087 | 0.727 | 0.213 | 0.611 | 0.128 | 0.697 |
| AttCAT | 0.078 | 0.740 | 0.063 | 0.747 | 0.205 | 0.623 | 0.119 | 0.703 |
| TIS | 0.104 | 0.719 | 0.082 | 0.737 | 0.252 | 0.562 | 0.135 | 0.691 |
| Contrast-CAT | **0.058** | **0.757** | **0.048** | **0.759** | **0.147** | **0.669** | **0.047** | **0.775** |

Table 1: AUC values from the faithfulness evaluation, with (A) showing results under the MoRF (Most Relevant First) setting and (B) showing results under the LeRF (Least Relevant First) setting. The best and second-best results are highlighted in bold and underlined, respectively. The arrows mean that ↑: higher is better, and ↓: lower is better.

## 5.1 FAITHFULNESS OF ATTRIBUTION

Figure 3 illustrates the AOPC and LOdds values for attribution maps generated by each competing method, evaluated at various top-$k\%$ thresholds where $k$ is increased by 10 within the range of $[10, 90]$. Table 1 provides the corresponding AUC values. Note that Figure 3 presents results for the MoRF setting only, while Table 1 includes results for both MoRF and LeRF settings (see supplementary material for LeRF results related to Figure 3). Through this evaluation, we can analyze the overall characteristics of an attribution map in terms of relevance scores of different threshold levels.

The trends in Figure 3 reveal that our method, Contrast-CAT, consistently maintains faithful attribution quality across all threshold levels and datasets compared to other methods. Table 1 further supports this, showing that Contrast-CAT consistently achieves top-1 attribution quality under both MoRF and LeRF settings. Specifically, compared to the second-best cases, Contrast-CAT shows average improvements in AUC values of AOPC and LOdds under the MoRF setting by $\times 1.30$ and $\times 2.25$, respectively. For the LeRF setting, Contrast-CAT shows average improvements in AUC values of AOPC and LOdds by $\times 1.34$ and $\times 1.03$, respectively.

## 5.2 QUALITATIVE EVALUATION

Figure 4 illustrates the attribution maps generated by Contrast-CAT, TIS, and AttCAT, the top-3 ranked methods in our faithfulness evaluation, conducted under the MoRF setting (Table 1, (A) MoRF). The examples provided are from the SST2 dataset. For ease of interpretation, only tokens

| | Class : Negative | Class : Positive |
|---|---|---|
| **Input** | the movie fails to live up to the sum of its parts. | rare birds has more than enough charm to make it memorable. |
| **Contrast-CAT** | the movie fails to live up to the sum of its parts . | rare birds has more than enough charm to make it memorable . |
| **AttCAT** | the movie fails to live up to the sum of its parts . | rare birds has more than enough charm to make it memorable . |
| **TIS** | the movie fails to live up to the sum of its parts . | rare birds has more than enough charm to make it memorable . |
| **Input** | my reaction in a word : disappointment. | a warm, funny, engaging film. |
| **Contrast-CAT** | my reaction in a word : disappointment . | a warm , funny , engaging , film . |
| **AttCAT** | my reaction in a word : disappointment . | a warm , funny , engaging , film . |
| **TIS** | my reaction in a word : disappointment . | a warm , funny , engaging , film . |

Figure 4: Qualitative comparison of attribution quality. Relevance scores are shown with color shades: red for the highest importance, followed by orange.

| Dataset | Amazon | | Yelp | | SST2 | | IMDB | |
|---|---|---|---|---|---|---|---|---|
| Reference | AOPC↑ | LOdds↓ | AOPC↑ | LOdds↓ | AOPC↑ | LOdds↓ | AOPC↑ | LOdds↓ |
| Random | 0.513 | 0.306 | 0.496 | 0.323 | 0.433 | 0.398 | 0.634 | 0.213 |
| Same | 0.144 | 0.667 | 0.159 | 0.650 | 0.089 | 0.728 | 0.124 | 0.614 |
| Contrasting | **0.703** | **0.117** | **0.687** | **0.131** | **0.654** | **0.157** | **0.738** | **0.101** |

Table 2: The effect of our activation contrasting approach, measured under the MoRF (Most Relevant First) setting. 'Random' uses randomly selected references (the mean values over 30 repetitions are reported), 'Same' uses references from the same class as the target, and 'Contrasting' refers to the suggested Contrast-CAT. The best results are in boldface.

with relevance scores exceeding 0.5 are highlighted. As shown in the left side of Figure 4, Contrast-CAT effectively identifies relevant tokens related to the predicted class, such as 'fails' or 'disappointment' for the negative prediction cases. For a positive prediction, in the input phrase 'rare birds have more than enough charm to make it memorable.', Contrast-CAT highlights 'enough' and 'charm' as the most relevant tokens, with 'than', 'make', 'more', and 'memorable' following in relevance. In contrast, AttCAT focuses only on 'enough' and 'memorable', missing 'charm' and 'more', while TIS identifies 'to' as the most relevant token. In another example, 'a warm, funny, engaging film.', Contrast-CAT precisely identifies 'warm', 'funny', and 'engaging' as key tokens, whereas the other methods either highlight irrelevant tokens like commas or fail to highlight any relevant tokens.

## 5.3 THE EFFECT OF ACTIVATION CONTRASTING

To evaluate the effect of our Contrast-CAT's activation contrasting, we compared the attribution quality of different versions of Contrast-CAT: the 'Random' version uses randomly selected references from individual training datasets instead of what had been outlined in Section 4.1, and the 'Same' version uses references of the same class as the target instead of different classes. The 'Same' version contrasts with our method, which leverages activations from different classes as contrastive references to reduce class-irrelevant features in the target activations.

Table 2 presents AUC values of each version of Contrast-CAT, where the suggested Contrast-CAT is denoted by 'Contrasting'. The attribution quality is the worst with 'Same' and the best with 'Contrasting', which indicates that the proposed activation contrasting effectively reduces class-irrelevant features in the activations, thereby helping to generate high-quality attribution maps.

## 5.4 THE EFFECT OF USING MULTIPLE LAYERS

Panel (A) of Figure 5 demonstrates the effect of using multiple layers to improve the attribution quality of Contrast-CAT. The figure shows the average AUC values of AOPC and LOdds across datasets, measured under the MoRF setting.

The results in panel (A) of Figure 5 indicate that the attribution quality improves as the number of layers increases, with the best performance achieved when all layers are used, as indicated by the higher AOPC and lower LOdds values. Specifically, there is a $\times 1.52$ improvement in AOPC and $\times 3.05$ improvement in LOdds when using all layers compared to using only the penultimate layer.

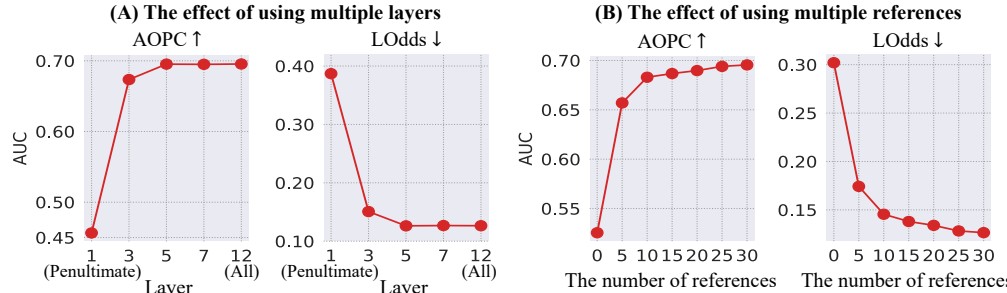

Figure 5: Comparison of Contrast-CAT's attribution quality measured under the MoRF (Most Relevant First) setting: (A) as the number of layers used to generate attribution maps increases from the penultimate layer to all layers, and (B) as the number of references used for multiple contrasts increases from 0 to 30.

The AOPC and LOdds values tend to saturate when we use three or more layers but continue to increase as the number increases.

### 5.5 THE EFFECT OF MULTIPLE CONTRASTS

Panel (B) of Figure 5 illustrates the impact of increasing the number of references for multiple contrasts in Contrast-CAT on attribution quality. The figure presents the average AUC values for AOPC and LOdds across datasets, measured under the MoRF setting.

The AOPC metric shows a sharp improvement as the number of references increases from 0 to 5, with the AUC rising from around 0.55 to 0.68. After 5 references, the AUC continues to increase, stabilizing between 25 and 30 references, plateauing around 0.70. In contrast, the LOdds metric exhibits a sharp decline as the number of references increases, starting at approximately 0.30 and dropping steadily, stabilizing around 0.10 after 10 references and reaching its minimum at 30 references. These results demonstrate that increasing the number of references improves attribution quality, with the best performance observed at 30 references, which we used in our experiment.

### 5.6 CONFIDENCE OF ATTRIBUTION

If an attribution method consistently generates similar attribution maps regardless of the model's prediction, the confidence of such a method will be questionable. Therefore, we conducted the confidence evaluation of the attribution methods employing the Kendall-$\tau$ rank correlation (Kendall, 1948), which is a statistical measure used to assess the similarity between two data by comparing the ranking order of their respective values. We compute an averaged rank correlation:

$$\frac{1}{N} \sum_{i=1}^{N} \text{Kendall-}\tau(P_i^c, P_i^{\hat{c}}),$$

where $P_i^c$ is an array of token indices in descending order of relevance scores for class $c$ in an attribution map, $P_i^{\hat{c}}$ is a similar array but for the class $\hat{c} \neq c$, and $N$ is the total number of data points used for testing. For the choice of $\hat{c}$, we followed the settings of AttCAT as detailed in their open-source implementation, where the class immediately following the class $c$ was chosen.

| Method | Dataset | | | |
|---|---|---|---|---|
| | Amazon | Yelp | SST2 | IMDB |
| RawAtt | 1.00 | 1.00 | 1.00 | 1.00 |
| Rollout | 1.00 | 1.00 | 1.00 | 1.00 |
| Att-grads | < 0.05 | < 0.05 | < 0.05 | < 0.05 |
| Att×Att-grads | < 0.05 | < 0.05 | < 0.05 | < 0.05 |
| Grad-SAM | 0.158 | 0.138 | 0.282 | 0.084 |
| Full LRP | 0.732 | 0.629 | 0.712 | 0.533 |
| Partial LRP | 0.952 | 0.924 | 0.957 | 0.859 |
| TransAtt | 0.153 | 0.135 | 0.342 | 0.061 |
| CAT | < 0.05 | < 0.05 | < 0.05 | < 0.05 |
| AttCAT | < 0.05 | < 0.05 | < 0.05 | < 0.05 |
| TIS | < 0.05 | < 0.05 | < 0.05 | < 0.05 |
| Contrast-CAT | < 0.05 | < 0.05 | < 0.05 | < 0.05 |

Table 3: The results of the confidence evaluation, showing averaged rank correlation values. The values below 0.05 (marked in gray) indicate that attributions tend to be class-distinct, as desired.

If an attribution method assigns relevance scores to tokens in distinct orders for different class predictions of the inspected model, the rank correlation is expected to be low. Table 3 presents the average rank correlation for various attribution methods tested across different datasets. The cases with average rank correlation values under 0.05 are marked as '$< 0.05$' and highlighted: these are the cases where the attribution methods seem to work soundly – our Contrast-CAT seems to pass the test, along with Att-grads, Att×Att-grads, CAT, AttCAT and TIS. In contrast, attribution methods such as RawAtt, Rollout, and Partial LRP showed values near 1.0 consistently over the datasets, suggesting that these methods have issues generating distinct attribution over different class outcomes.

## 6 CONCLUSION

In this work, we reported that activation-based attribution methods for interpreting transformer-based text classification models may incorporate class-irrelevant features into attribution maps, potentially leading to a degradation in attribution quality. To address this challenge, we introduced Contrast-CAT, a novel activation-based attribution method that leverages activation contrasting to reduce class-irrelevant features within activations, thereby generating high-quality token-level attribution maps. Our extensive experiments demonstrated that Contrast-CAT significantly outperforms state-of-the-art methods in terms of faithfulness, as measured by AOPC and LOdds metrics, under both MoRF and LeRF settings.

Despite its effectiveness, Contrast-CAT requires reference points whose activations must be available during the creation of attribution maps. We have minimized the computational overhead using a pre-built reference library; however, it will require larger storage as the number of classes and the size of activation maps increase. To address this, we plan to explore replacing the reference activations with alternative tensors that can be computed and stored at a lower cost, ideally without relying on training data in future work.

Nevertheless, given the growing need to interpret AI models' decisions to ensure their safety, security, and trustworthiness, we believe that Contrast-CAT serves as a meaningful advancement in improving the interpretability and transparency of transformer-based models.

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

# APPENDIX

In this section, we provide implementation details and additional experimental results.

## A  ALGORITHM AND IMPLEMENTATION

---

**Algorithm 1** Contrast-CAT

---

**Input**: An input token sequence $x$ with length $T$, a target class $c$, its prediction score $f_c(\cdot)$, and the activation $A$.
**Input**: Lib$_c$, a list of reference activations for the class $c$.
**Parameter**: Maximum number of references $K$.

1: Initialize $I$ as an empty array of length $K$.
2: **for** $r \leftarrow 1$ to $K$ **do**
3:     $R \leftarrow \text{Lib}_c[r]$.
4:     **for** $i \leftarrow 1$ to $T$ **do**
5:         $\hat{\alpha}_i^\ell \leftarrow \frac{1}{HT} \sum_{h=1}^H \sum_{j=1}^T \alpha_{i,j}^{\ell,h}$.
6:         $I_i^\ell \leftarrow \hat{\alpha}_i^\ell \sum_{j=1}^n \left( \frac{\partial f_c(x)}{\partial A^\ell} \odot (A_i^\ell - R_i^\ell) \right)_j$.
7:     **end for**
8:     $I[r] \leftarrow \sum_\ell I^\ell$.
9: **end for**
10: **for** each $r$ from 1 to $K$ **do** {Parallel execution}
11:     $\hat{x}, I_r \leftarrow x, I[r]$.
12:     **for** from most to least relevant according to $I_r$ **do**
13:         Remove the token at index $i$ from $\hat{x}$.
14:         $S[r,i] \leftarrow f_c(x) - f_c(\hat{x})$.
15:     **end for**
16:     $S[r] \leftarrow \frac{1}{T} \sum_i S[r,i]$.
17: **end for**
18: $D \leftarrow \{\text{indices } r \text{ for which } S[r] \geq \rho\}$.
19: If $D = \emptyset$, $D \leftarrow \{1, \ldots, K\}$.
20: $I_{\text{Contrast-CAT}} \leftarrow \frac{1}{|D|} \sum_{r \in D} I[r]$
21: **return** $I_{\text{Contrast-CAT}}$

---

We conducted our experiments using several libraries, including Python v3.7.4, PyTorch v1.9.1, scikit-learn v0.22.2, Hugging Face Hub v0.14.1, Transformers v4.29.1, OpenCV-Python v4.2.0.32, and NumPy v1.17.4. We set the random seed across all libraries to 41.

The detailed procedures of Contrast-CAT are outlined in Algorithm 1.

## B  DATASETS

In our experiments, we used five publicly available NLP datasets for text classification tasks: Amazon Polarity (Zhang et al., 2015), Yelp Polarity (Zhang et al., 2015), SST2 (Socher et al., 2013), IMDB (Maas et al., 2011), and AgNews (Del Corso et al., 2005). Details on the train/test set split for each dataset are provided in Table 4.

| Dataset | Amazon | Yelp | SST2 | IMDB | AgNews |
|---------|--------|------|------|------|--------|
| Trainset | 3600000 | 560000 | 67349 | 25000 | 120000 |
| Testset | 400000 | 38000 | 1821 | 25000 | 7600 |

Table 4: The number of samples in the train/test splits for the five datasets used in our experiments.

| Model | Dataset | | | | |
|---|---|---|---|---|---|
| | Amazon | Yelp | SST2 | IMDB | AgNews |
| BERT$_{base}$ | 0.946 | 0.956 | 0.924 | 0.930 | 0.941 |
| DistilBERT | 0.945 | 0.962 | 0.891 | 0.928 | 0.947 |
| RoBERTa | 0.953 | 0.982 | 0.940 | 0.953 | 0.947 |
| GPT-2 | 0.968 | 0.977 | 0.921 | 0.877 | 0.949 |

Table 5: Test accuracy of transformer-based text classification models used in our experiments.

## C    TRANSFORMER MODELS

We conducted our experiments using four types of transformer-based models: BERT$_{base}$ (Devlin et al., 2019), DistilBERT (Sanh et al., 2019), RoBERTa (Liu et al., 2019), and GPT-2 (Radford et al., 2019). We used pre-trained versions of these models from Hugging Face (Wolf et al., 2019) for the datasets used in our experiments. Table 5 presents the accuracies of each pre-trained model on the five datasets used in our experiments.

The pre-trained **BERT$_{base}$** models we used are sourced from:

| Amazon | `https://huggingface.co/fabriceyhc/`
`bert-base-uncased-amazon_polarity` |
|---|---|
| Yelp | `https://huggingface.co/fabriceyhc/`
`bert-base-uncased-yelp_polarity` |
| SST2 | `https://huggingface.co/textattack/`
`bert-base-uncased-SST-2` |
| IMDB | `https://huggingface.co/fabriceyhc/`
`bert-base-uncased-imdb` |
| AgNews | `https://huggingface.co/nateraw/`
`bert-base-uncased-ag-news` |

The pre-trained **DistilBERT** models we used are sourced from:

| Amazon | `https://huggingface.co/AdamCodd/`
`distilbert-base-uncased-finetuned-sentiment-amazon` |
|---|---|
| Yelp | `https://huggingface.co/randellcotta/`
`distilbert-base-uncased-finetuned-yelp-polarity` |
| SST2 | `https://huggingface.co/assemblyai/`
`distilbert-base-uncased-sst2` |
| IMDB | `https://huggingface.co/lvwerra/distilbert-imdb` |
| AgNews | `https://huggingface.co/andi611/`
`distilbert-base-uncased-ner-agnews` |

The pre-trained **RoBERTa** models we used are sourced from:

| Amazon | `https://huggingface.co/pig4431/amazonPolarity_`
`roBERTa_5E` |
|---|---|
| Yelp | `https://huggingface.co/VictorSanh/`
`roberta-base-finetuned-yelp-polarity` |
| SST2 | `https://huggingface.co/textattack/`
`roberta-base-SST-2` |
| IMDB | `https://huggingface.co/textattack/`
`roberta-base-imdb` |
| AgNews | `https://huggingface.co/textattack/`
`roberta-base-ag-news` |

The pre-trained **GPT-2** models we used are sourced from:

| Amazon | `https://huggingface.co/ashok2216/`
`gpt2-amazon-sentiment-classifier-V1.0` |
|---|---|
| Yelp | `https://huggingface.co/utahnlp/yelp_polarity_gpt2_`
`seed-2` |
| SST2 | `https://huggingface.co/michelecafagna26/`
`gpt2-medium-finetuned-sst2-sentiment` |
| IMDB | `https://huggingface.co/mnoukhov/`
`gpt2-imdb-sentiment-classifier` |
| AgNews | `https://huggingface.co/xinzhel/gpt2-ag-news` |

## D  FAITHFULNESS OF ATTRIBUTION

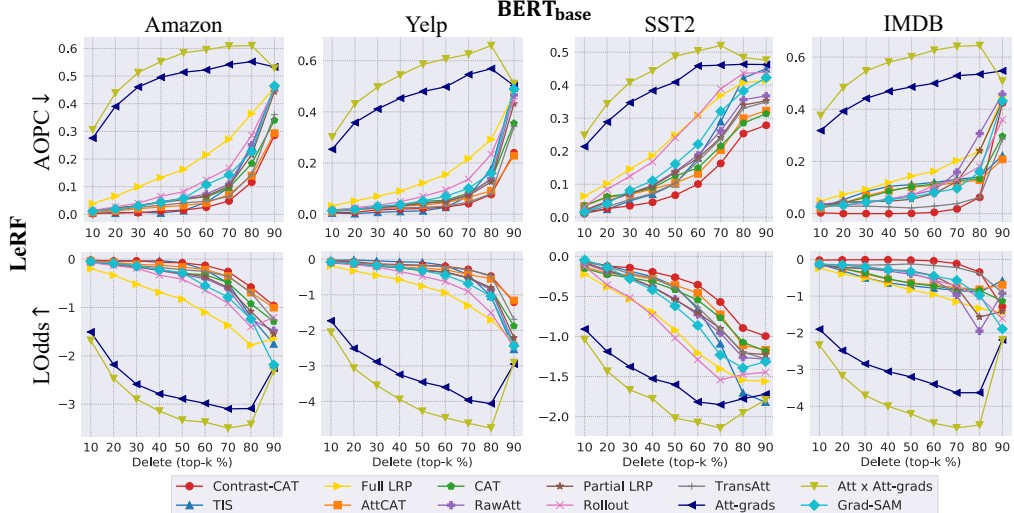

Figure 6: Quantitative comparison of the faithfulness evaluation of Contrast-CAT and other attribution methods, measured under the LeRF (Least Relevant First) setting.

**Additional Experimental Results for the BERT$_{base}$ Model**  Figure 6 shows the faithfulness evaluation results under the LeRF setting, corresponding to the results labeled as (B) LeRF in Table 1 of our main manuscript. Table 6 presents the faithfulness evaluation results of attribution methods on the AgNews dataset using the BERT$_{base}$ model, following the settings outlined in Section 5.1.

As shown in Table 6, Contrast-CAT demonstrates consistently superior attribution quality on the AgNews dataset compared to other competing methods, similar to the results in Table 1.

**Experimental Results for Other Models**  We conducted the faithfulness evaluation of attribution methods, detailed in Section 5.1, using the DistilBERT (Sanh et al., 2019), RoBERTa (Liu et al., 2019), and GPT-2 (Radford et al., 2019) models. In these experiments, we compared Contrast-CAT with five different attribution methods: RawAtt and Rollout (attention-based methods), and CAT, AttCAT, and TIS (activation-based methods).

Figure 7 and Table 7 present the results for the DistilBERT model, while Figure 8 and Table 8 show the results for the RoBERTa model, and Figure 9 and Table 9 display the results for the GPT-2 model. The results for TIS are omitted from Figure 9 and marked as N/A in Table 9 since it is not applicable to the GPT-2 model. Table 10 shows the results on the AgNews dataset for the DistilBERT, RoBERTa, and GPT-2 models.

The results consistently demonstrate the superior attribution quality of Contrast-CAT across different datasets and models. Specifically, for the DistilBERT model, average improvements across

| Setting | MoRF
(Most Relevant First) | | LeRF
(Least Relevant First) | |
|---|---|---|---|---|
| Method | AOPC↑ | LOdds↓ | AOPC↓ | LOdds↑ |
| RawAtt | 0.268 | 0.580 | 0.152 | 0.663 |
| Rollout | 0.300 | 0.532 | 0.184 | 0.639 |
| Att-grads | 0.099 | 0.728 | 0.331 | 0.461 |
| Att×Att-grads | 0.084 | 0.739 | 0.379 | 0.394 |
| Grad-SAM | 0.270 | 0.578 | 0.180 | 0.632 |
| Full LRP | 0.234 | 0.604 | 0.199 | 0.623 |
| Partial LRP | 0.294 | 0.555 | 0.135 | 0.681 |
| TransAtt | 0.347 | 0.499 | 0.105 | 0.714 |
| CAT | 0.273 | 0.556 | 0.137 | 0.680 |
| AttCAT | 0.289 | 0.536 | 0.126 | 0.692 |
| TIS | 0.354 | 0.473 | 0.143 | 0.674 |
| Contrast-CAT | **0.434** | **0.363** | **0.093** | **0.723** |

Table 6: AUC values for the faithfulness evaluation of attribution methods using the **BERT$_{base}$** model on the AgNews dataset under the MoRF (Most Relevant First) and LeRF (Least Relevant First) settings. The best and the second-best cases are in boldface and underlined, respectively.

different datasets are ×1.31 in AOPC and ×2.39 in LOdds compared to the second-best methods under the MoRF setting. Under the LeRF setting, Contrast-CAT shows average improvements in AUC values for AOPC and LOdds by ×1.39 and ×1.07, respectively. For the RoBERTa model, the average improvements are ×1.61 in AOPC and ×2.97 in LOdds under the MoRF setting, with AUC improvements of ×2.07 and ×1.12 in AOPC and LOdds, respectively, under the LeRF setting. Similarly, for the GPT-2 model, the average improvements across datasets are ×2.78 in AOPC and ×3.37 in LOdds under the MoRF setting. For the LeRF setting, Contrast-CAT demonstrates average improvements of ×3.80 in AOPC and ×1.39 in LOdds.

These results align with those presented in Figure 3 and Table 1 of our main manuscript, further validating Contrast-CAT's superiority in generating faithful attribution maps.

| (A) MoRF (Most Relevant First) | | | | | | | | |
|---|---|---|---|---|---|---|---|---|
| Dataset | Amazon | | Yelp | | SST2 | | IMDB | |
| Method | AOPC↑ | LOdds↓ | AOPC↑ | LOdds↓ | AOPC↑ | LOdds↓ | AOPC↑ | LOdds↓ |
| RawAtt | 0.360 | 0.557 | 0.306 | 0.618 | 0.363 | 0.531 | 0.172 | 0.729 |
| Rollout | 0.307 | 0.638 | 0.242 | 0.676 | 0.322 | 0.587 | 0.231 | 0.700 |
| CAT | 0.521 | 0.361 | 0.528 | 0.334 | 0.469 | 0.392 | 0.625 | 0.235 |
| AttCAT | 0.532 | 0.341 | 0.570 | 0.278 | 0.480 | 0.376 | 0.638 | 0.217 |
| TIS | 0.436 | 0.448 | 0.406 | 0.476 | 0.394 | 0.467 | 0.428 | 0.487 |
| Contrast-CAT | **0.720** | **0.108** | **0.727** | **0.106** | **0.685** | **0.137** | **0.752** | **0.101** |
| (B) LeRF (Least Relevant First) | | | | | | | | |
| Dataset | Amazon | | Yelp | | SST2 | | IMDB | |
| Method | AOPC↓ | LOdds↑ | AOPC↓ | LOdds↑ | AOPC↓ | LOdds↑ | AOPC↓ | LOdds↑ |
| RawAtt | 0.174 | 0.626 | 0.122 | 0.649 | 0.283 | 0.508 | 0.121 | 0.673 |
| Rollout | 0.181 | 0.606 | 0.112 | 0.655 | 0.328 | 0.429 | 0.090 | 0.706 |
| CAT | 0.119 | 0.678 | 0.065 | 0.708 | 0.248 | 0.536 | 0.028 | 0.773 |
| AttCAT | 0.098 | 0.703 | 0.028 | 0.764 | 0.234 | 0.549 | 0.016 | 0.787 |
| TIS | 0.162 | 0.637 | 0.113 | 0.669 | 0.315 | 0.478 | 0.089 | 0.708 |
| Contrast-CAT | **0.068** | **0.737** | **0.020** | **0.779** | **0.142** | **0.669** | **0.015** | **0.788** |

Table 7: AUC values of the faithfulness evaluation conducted on the **DistilBERT** model. The best and the second-best cases are in boldface and underlined, respectively.

| (A) MoRF (Most Relevant First) | | | | | | | | |
|---|---|---|---|---|---|---|---|---|
| Dataset | Amazon | | Yelp | | SST2 | | IMDB | |
| Method | AOPC↑ | LOdds↓ | AOPC↑ | LOdds↓ | AOPC↑ | LOdds↓ | AOPC↑ | LOdds↓ |
| RawAtt | 0.245 | 0.615 | 0.164 | 0.713 | 0.260 | 0.619 | 0.272 | 0.676 |
| Rollout | 0.188 | 0.660 | 0.153 | 0.717 | 0.195 | 0.653 | 0.274 | 0.657 |
| CAT | 0.287 | 0.557 | 0.357 | 0.526 | 0.461 | 0.410 | 0.452 | 0.464 |
| AttCAT | 0.274 | 0.568 | 0.347 | 0.532 | 0.454 | 0.416 | 0.449 | 0.467 |
| TIS | 0.354 | 0.503 | 0.394 | 0.503 | 0.524 | 0.372 | 0.520 | 0.411 |
| Contrast-CAT | **0.688** | **0.140** | **0.684** | **0.160** | **0.686** | **0.160** | **0.738** | **0.131** |

| (B) LeRF (Least Relevant First) | | | | | | | | |
|---|---|---|---|---|---|---|---|---|
| Dataset | Amazon | | Yelp | | SST2 | | IMDB | |
| Method | AOPC↓ | LOdds↑ | AOPC↓ | LOdds↑ | AOPC↓ | LOdds↑ | AOPC↓ | LOdds↑ |
| RawAtt | 0.218 | 0.586 | 0.177 | 0.581 | 0.323 | 0.457 | 0.157 | 0.556 |
| Rollout | 0.303 | 0.514 | 0.197 | 0.569 | 0.443 | 0.311 | 0.184 | 0.531 |
| CAT | 0.200 | 0.606 | 0.127 | 0.674 | 0.141 | 0.676 | 0.077 | 0.704 |
| AttCAT | 0.200 | 0.604 | 0.124 | 0.677 | 0.137 | 0.678 | 0.077 | 0.709 |
| TIS | 0.191 | 0.613 | 0.119 | 0.669 | 0.143 | 0.679 | 0.076 | 0.712 |
| Contrast-CAT | **0.065** | **0.741** | **0.052** | **0.771** | **0.085** | **0.738** | **0.053** | **0.749** |

Table 8: AUC values of the faithfulness evaluation conducted on the **RoBERTa** model. The best and the second-best cases are in boldface and underlined, respectively.

| (A) MoRF (Most Relevant First) | | | | | | | | |
|---|---|---|---|---|---|---|---|---|
| Dataset | Amazon | | Yelp | | SST2 | | IMDB | |
| Method | AOPC↑ | LOdds↓ | AOPC↑ | LOdds↓ | AOPC↑ | LOdds↓ | AOPC↑ | LOdds↓ |
| RawAtt | 0.385 | 0.622 | 0.138 | 0.690 | 0.303 | 0.420 | 0.163 | 0.699 |
| Rollout | 0.320 | 0.684 | 0.138 | 0.690 | 0.303 | 0.420 | 0.163 | 0.699 |
| CAT | 0.505 | 0.392 | 0.177 | 0.653 | 0.243 | 0.617 | 0.042 | 0.775 |
| AttCAT | 0.541 | 0.345 | 0.186 | 0.647 | 0.221 | 0.662 | 0.043 | 0.775 |
| TIS | N/A | N/A | N/A | N/A | N/A | N/A | N/A | N/A |
| Contrast-CAT | **0.744** | **0.136** | **0.617** | **0.188** | **0.636** | **0.188** | **0.706** | **0.132** |

| (B) LeRF (Least Relevant First) | | | | | | | | |
|---|---|---|---|---|---|---|---|---|
| Dataset | Amazon | | Yelp | | SST2 | | IMDB | |
| Method | AOPC↓ | LOdds↑ | AOPC↓ | LOdds↑ | AOPC↓ | LOdds↑ | AOPC↓ | LOdds↑ |
| RawAtt | 0.193 | 0.513 | 0.200 | 0.524 | 0.391 | 0.350 | 0.200 | 0.679 |
| Rollout | 0.215 | 0.472 | 0.200 | 0.524 | 0.391 | 0.350 | 0.200 | 0.679 |
| CAT | 0.164 | 0.584 | 0.247 | 0.434 | 0.492 | 0.321 | 0.703 | 0.199 |
| AttCAT | 0.129 | 0.646 | 0.216 | 0.488 | 0.506 | 0.359 | 0.679 | 0.231 |
| TIS | N/A | N/A | N/A | N/A | N/A | N/A | N/A | N/A |
| Contrast-CAT | **0.093** | **0.696** | **0.062** | **0.731** | **0.206** | **0.700** | **0.023** | **0.790** |

Table 9: AUC values of the faithfulness evaluation conducted on the **GPT-2** model. The best and the second-best cases are in boldface and underlined, respectively. N/A indicates that the method is not applicable to GPT-2.

| | (A) MoRF (Most Relevant First) | | | | | |
|---|---|---|---|---|---|---|
| Model | DistilBERT | | RoBERTa | | GPT-2 | |
| Method | AOPC↑ | LOdds↓ | AOPC↑ | LOdds↓ | AOPC↑ | LOdds↓ |
| RawAtt | 0.218 | 0.669 | 0.352 | 0.566 | 0.174 | 0.554 |
| Rollout | 0.316 | 0.620 | 0.181 | 0.673 | 0.174 | 0.554 |
| CAT | 0.344 | 0.492 | 0.333 | 0.530 | 0.174 | 0.557 |
| AttCAT | 0.345 | 0.487 | 0.330 | 0.540 | 0.176 | 0.575 |
| TIS | 0.323 | 0.556 | 0.413 | 0.456 | N/A | N/A |
| Contrast-CAT | **0.452** | **0.382** | **0.680** | **0.169** | **0.350** | **0.256** |
| | (B) LeRF (Least Relevant First) | | | | | |
| Model | DistilBERT | | RoBERTa | | GPT-2 | |
| Method | AOPC↓ | LOdds↑ | AOPC↓ | LOdds↑ | AOPC↓ | LOdds↑ |
| RawAtt | 0.225 | 0.609 | 0.188 | 0.580 | 0.178 | 0.484 |
| Rollout | 0.141 | 0.688 | 0.224 | 0.562 | 0.178 | 0.484 |
| CAT | 0.072 | **0.752** | 0.096 | 0.699 | 0.255 | 0.409 |
| AttCAT | **0.068** | **0.752** | 0.098 | 0.698 | 0.256 | 0.393 |
| TIS | 0.154 | 0.702 | 0.109 | 0.690 | N/A | N/A |
| Contrast-CAT | 0.072 | 0.746 | **0.061** | **0.742** | **0.161** | **0.588** |

Table 10: Faithfulness evaluation results of attribution methods conducted on the AgNews dataset using three models: **DistilBERT**, **RoBERTa**, and **GPT-2** under MoRF (Most Relevant First) and LeRF (Least Relevant First) settings.

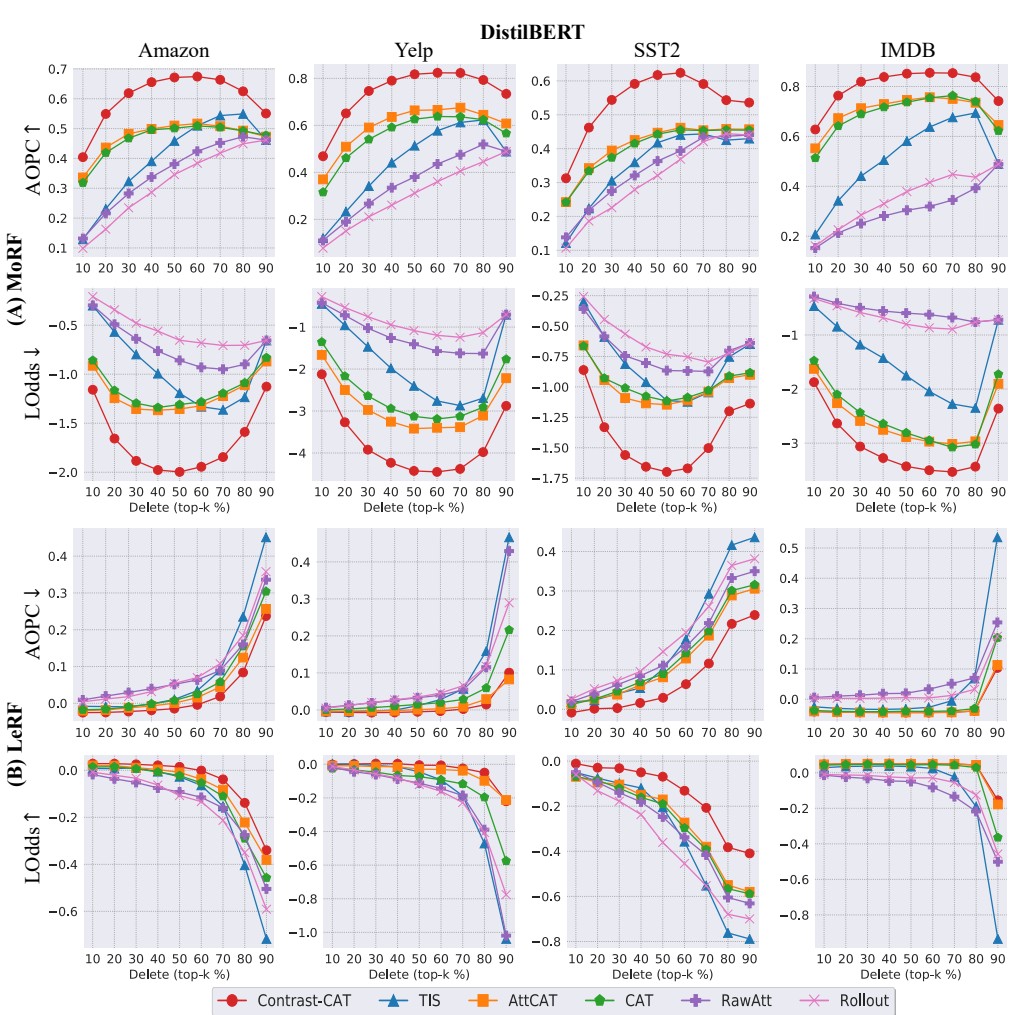

Figure 7: Faithfulness evaluation of our Contrast-CAT (red) and other attribution methods conducted on the **DistilBERT** model.

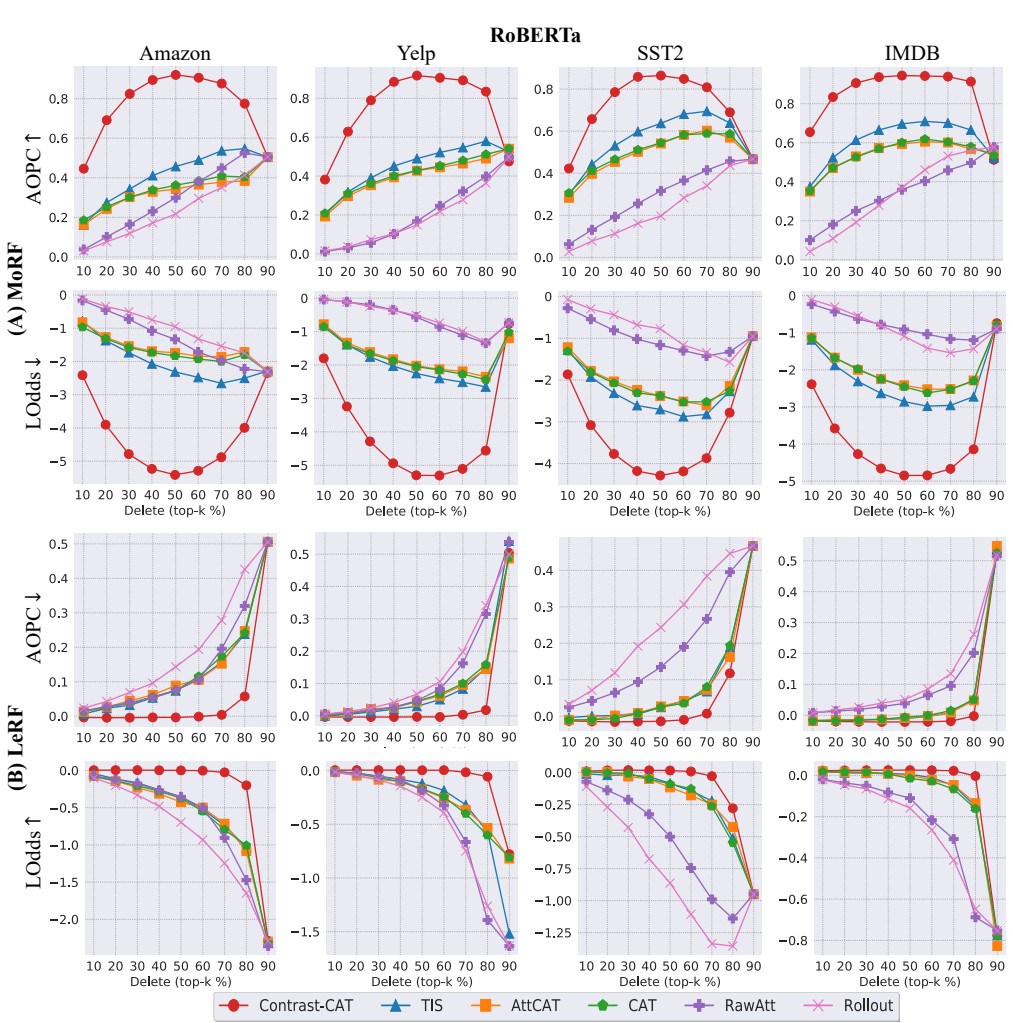

Figure 8: Faithfulness evaluation of our Contrast-CAT (red) and other attribution methods conducted on the **RoBERTa** model.

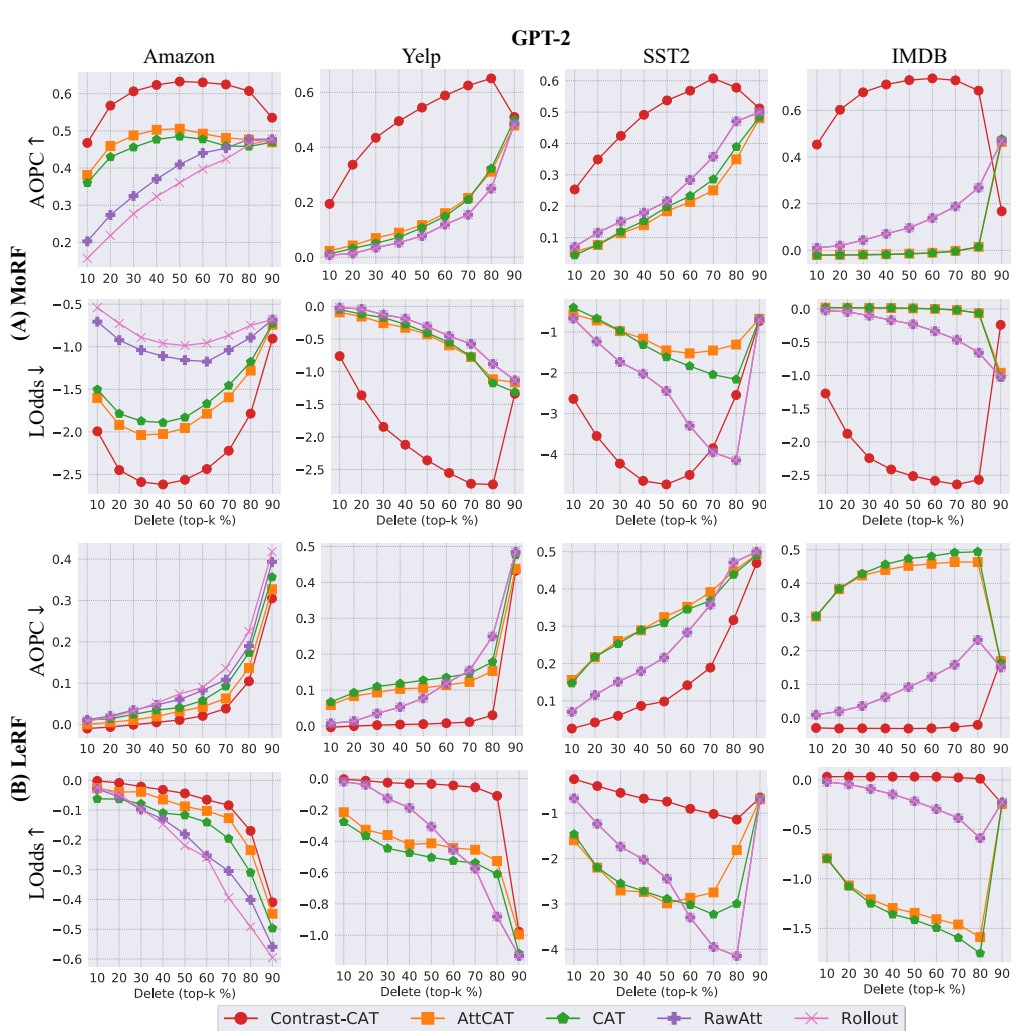

Figure 9: Faithfulness evaluation of our Contrast-CAT (red) and other attribution methods conducted on the **GPT-2** model.

# E  CONFIDENCE OF ATTRIBUTION

| Model | Dataset | RawAtt | Rollout | CAT | AttCAT | TIS | Contrast-CAT |
|---|---|---|---|---|---|---|---|
| DistilBERT | Amazon | 1.00 | 1.00 | $< 0.05$ | $< 0.05$ | $< 0.05$ | $< 0.05$ |
| | Yelp | 1.00 | 1.00 | $< 0.05$ | $< 0.05$ | $< 0.05$ | $< 0.05$ |
| | SST2 | 1.00 | 1.00 | $< 0.05$ | $< 0.05$ | $< 0.05$ | $< 0.05$ |
| | IMDB | 1.00 | 1.00 | $< 0.05$ | $< 0.05$ | $< 0.05$ | $< 0.05$ |
| | AgNews | 1.00 | 1.00 | 0.069 | $< 0.05$ | $< 0.05$ | $< 0.05$ |
| RoBERTa | Amazon | 1.00 | 1.00 | $< 0.05$ | $< 0.05$ | $< 0.05$ | $< 0.05$ |
| | Yelp | 1.00 | 1.00 | $< 0.05$ | $< 0.05$ | $< 0.05$ | $< 0.05$ |
| | SST2 | 1.00 | 1.00 | $< 0.05$ | $< 0.05$ | $< 0.05$ | $< 0.05$ |
| | IMDB | 1.00 | 1.00 | $< 0.05$ | $< 0.05$ | $< 0.05$ | $< 0.05$ |
| | AgNews | 1.00 | 1.00 | 0.050 | 0.054 | $< 0.05$ | $< 0.05$ |
| GPT-2 | Amazon | 1.00 | 1.00 | $< 0.05$ | $< 0.05$ | N/A | $< 0.05$ |
| | Yelp | 1.00 | 1.00 | $< 0.05$ | $< 0.05$ | N/A | $< 0.05$ |
| | SST2 | 1.00 | 1.00 | $< 0.05$ | $< 0.05$ | N/A | $< 0.05$ |
| | IMDB | 1.00 | 1.00 | $< 0.05$ | $< 0.05$ | N/A | $< 0.05$ |
| | AgNews | 1.00 | 1.00 | $< 0.05$ | 0.068 | N/A | $< 0.05$ |

Table 11: The results of confidence evaluation conducted on the **DistilBERT**, **RoBERTa**, and **GPT-2** models. Values below $0.05$ are marked in gray. N/A indicates that the method is not applicable to the given model.

Table 11 presents the confidence evaluation results for various attribution methods conducted on the DistilBERT, RoBERTa, and GPT-2 models. The results show that Contrast-CAT consistently achieves average rank correlation values below $0.05$ across all datasets and models used, suggesting that the attributions generated by Contrast-CAT tend to be class-distinct as desired.

# F  ACTIVATION VISUALIZATION

To demonstrate that Contrast-CAT's multiple contrasting, detailed in Section 4.2, effectively reduces class-irrelevant features in activations, we visualized activations from different layers of the $BERT_{base}$, DistilBERT, RoBERTa, and GPT-2 models, as shown in Figure 10. The 1st, 3rd, 5th, and 7th rows (odd-numbered rows) represent the original activations, while the 2nd, 4th, 6th, and 8th rows (even-numbered rows) show the activations after applying Contrast-CAT's multiple contrasting. In the case of $BERT_{base}$, RoBERTa, and GPT-2, the activations of layers 2, 4, 6, 8, and 10 were visualized. For DistilBERT, since it consists of only 6 layers, the activations of layers 1, 2, 3, 4, and 5 were visualized. Each point represents the averaged activation across tokens in an input token sequence, extracted from the corresponding layers. For visualization, the dimensionality of these averaged activations was reduced to two using Principal Component Analysis (F.R.S., 1901).

As illustrated in Figure 10, the original activations (odd-numbered rows in the figure) show poor separation between positive and negative classes. In contrast, after applying Contrast-CAT's multiple contrasting (even-numbered rows in the figure), the activations exhibit much clearer class separation across all layers. This enhanced separation highlights the effectiveness of Contrast-CAT in reducing class-irrelevant features within activations, thereby improving attribution quality by focusing on class-relevant features.

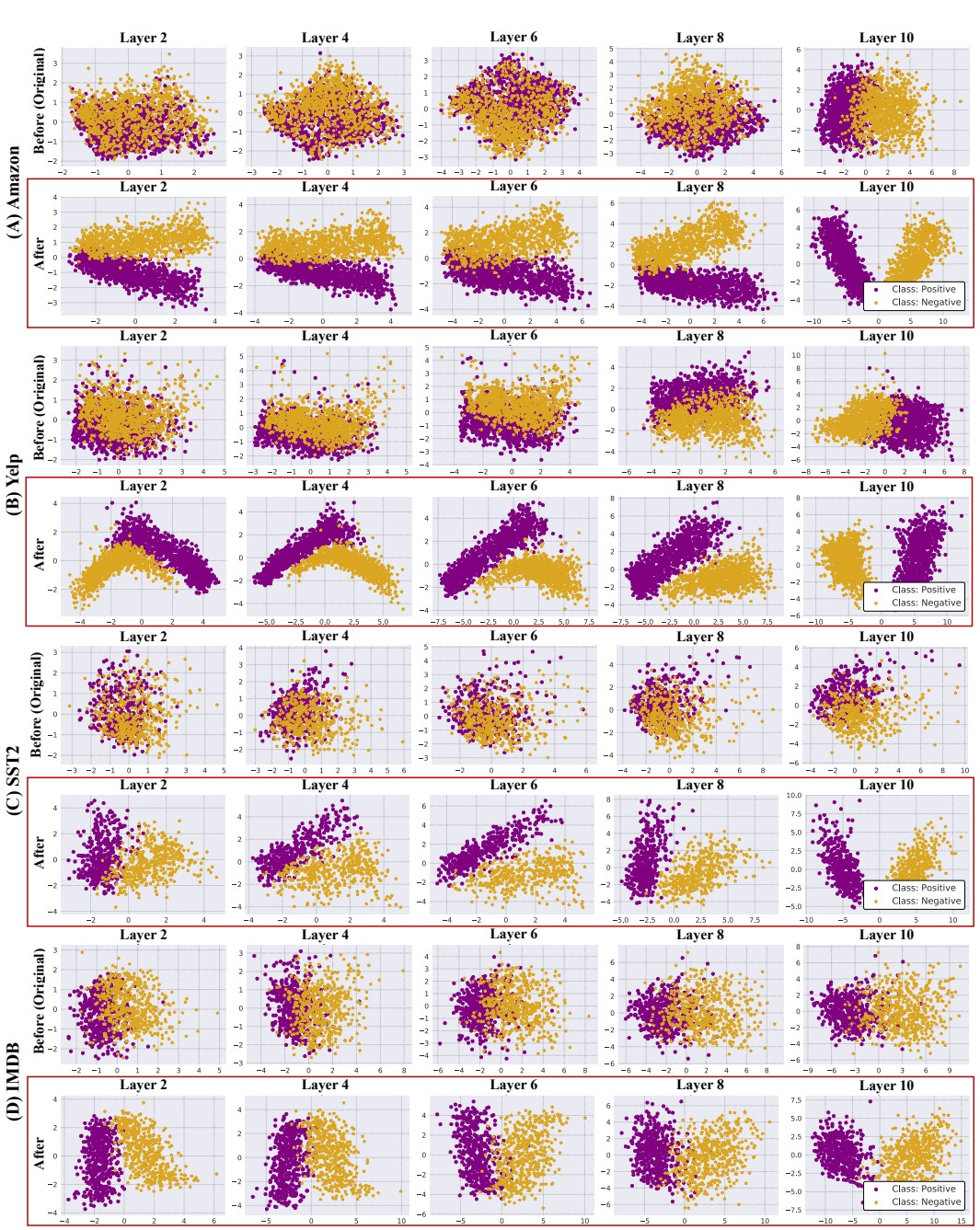

Figure 10: Visual representation of activations across five different layers of the **BERT**$_\text{base}$ model for four different datasets: (A) Amazon, (B) Yelp, (C) SST2, and (D) IMDB. Odd-numbered rows show activations before applying Contrast-CAT's multiple contrasting, and even-numbered rows (highlighted in a red box) show activations after applying Contrast-CAT's multiple contrasting. The colors represent classes: positive (yellow) and negative (purple). Principal Component Analysis is used to reduce the dimensionality of activations to two dimensions for visualization. The separation between positive (yellow) and negative (purple) classes becomes more distinct after applying Contrast-CAT's multiple contrasting.

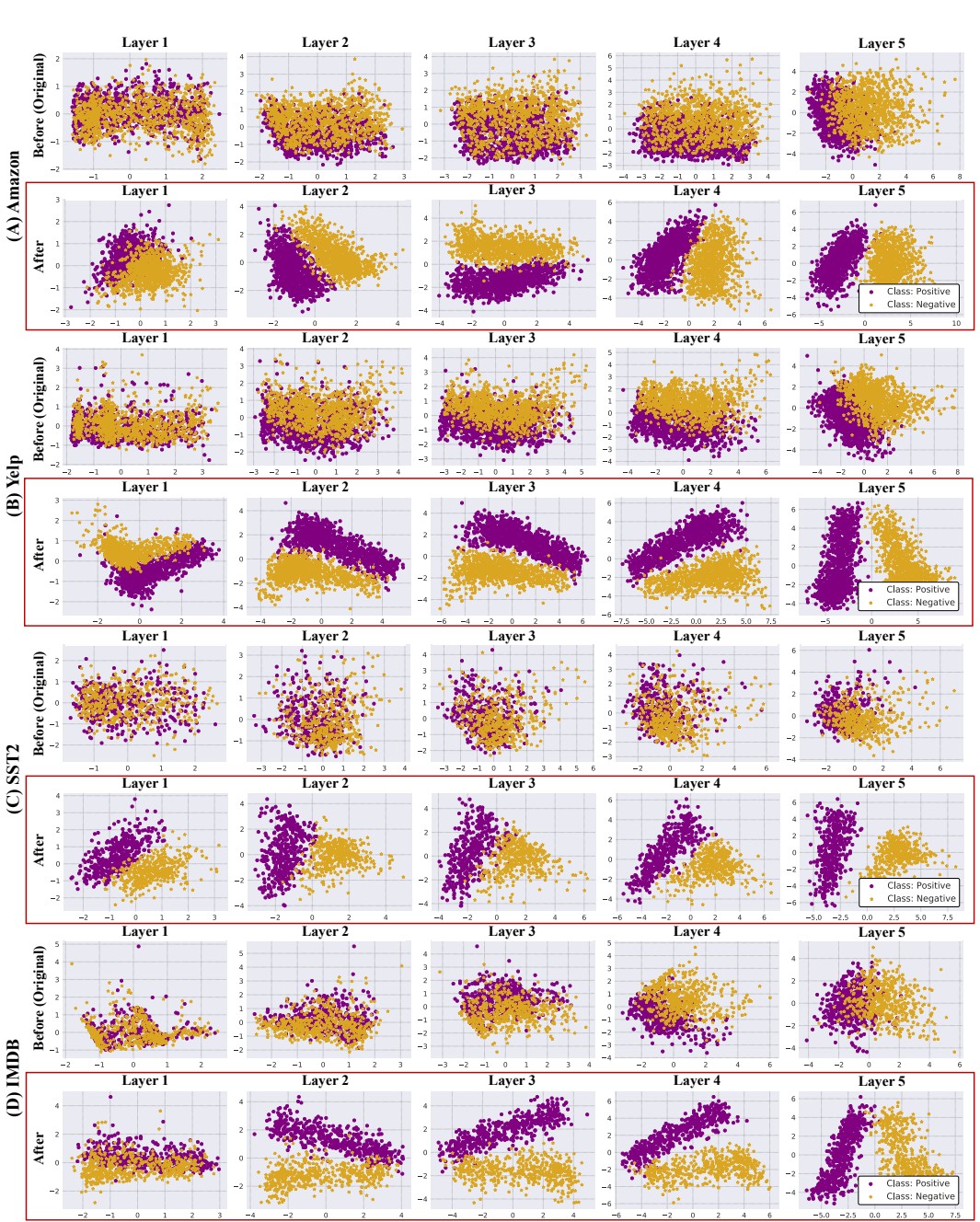

Figure 11: Visual representation of activations across five different layers of the **DistilBERT** model for four different datasets: (A) Amazon, (B) Yelp, (C) SST2, and (D) IMDB. Odd-numbered rows show activations before applying Contrast-CAT's multiple contrasting, and even-numbered rows (highlighted in a red box) show activations after applying Contrast-CAT's multiple contrasting. The colors represent classes: positive (yellow) and negative (purple). Principal Component Analysis is used to reduce the dimensionality of activations to two dimensions for visualization. The separation between positive (yellow) and negative (purple) classes becomes more distinct after applying Contrast-CAT's multiple contrasting.

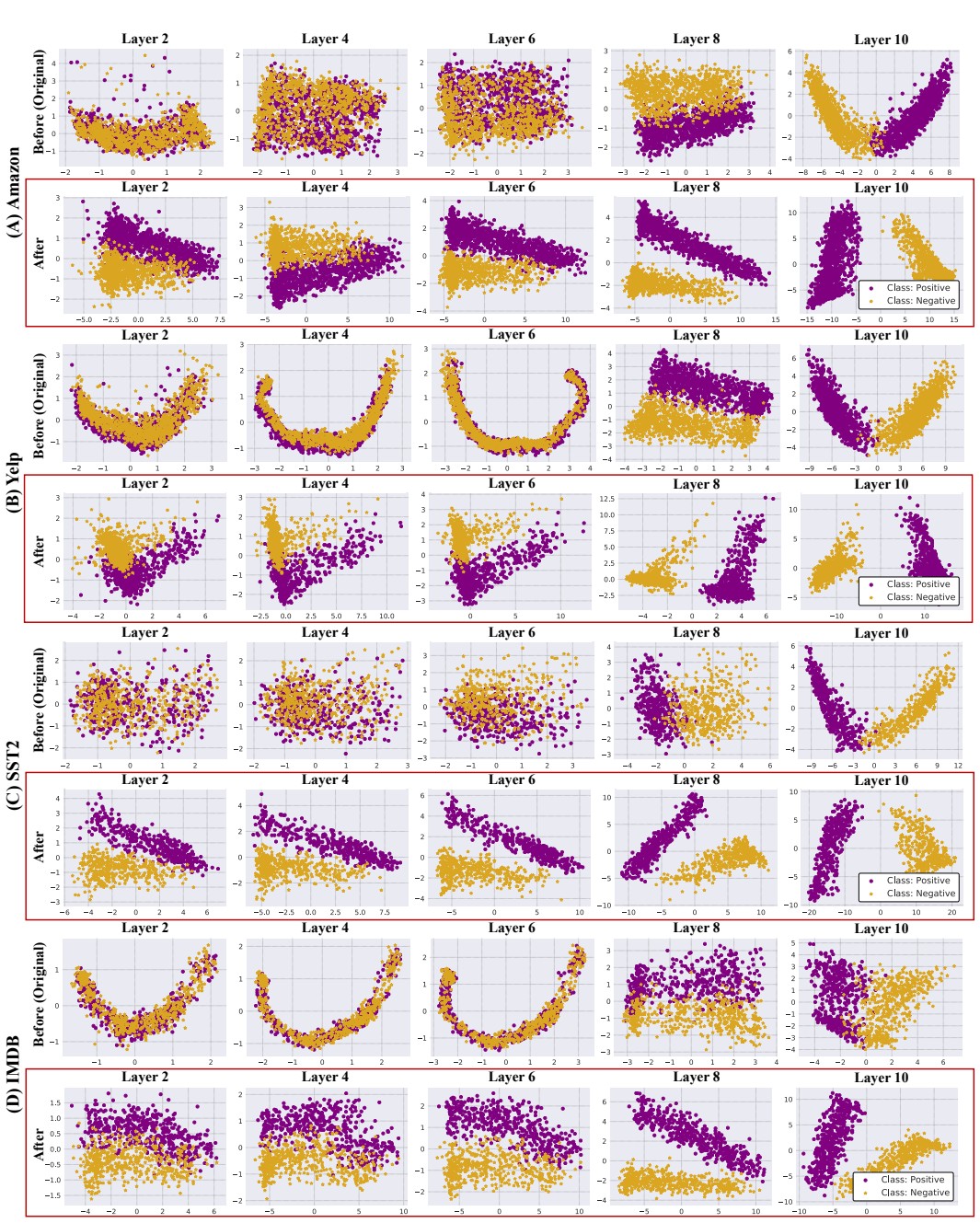

Figure 12: Visual representation of activations across five different layers of the **RoBERTa** model for four different datasets: (A) Amazon, (B) Yelp, (C) SST2, and (D) IMDB. Odd-numbered rows show activations before applying Contrast-CAT's multiple contrasting, and even-numbered rows (highlighted in a red box) show activations after applying Contrast-CAT's multiple contrasting. The colors represent classes: positive (yellow) and negative (purple). Principal Component Analysis is used to reduce the dimensionality of activations to two dimensions for visualization. The separation between positive (yellow) and negative (purple) classes becomes more distinct after applying Contrast-CAT's multiple contrasting.

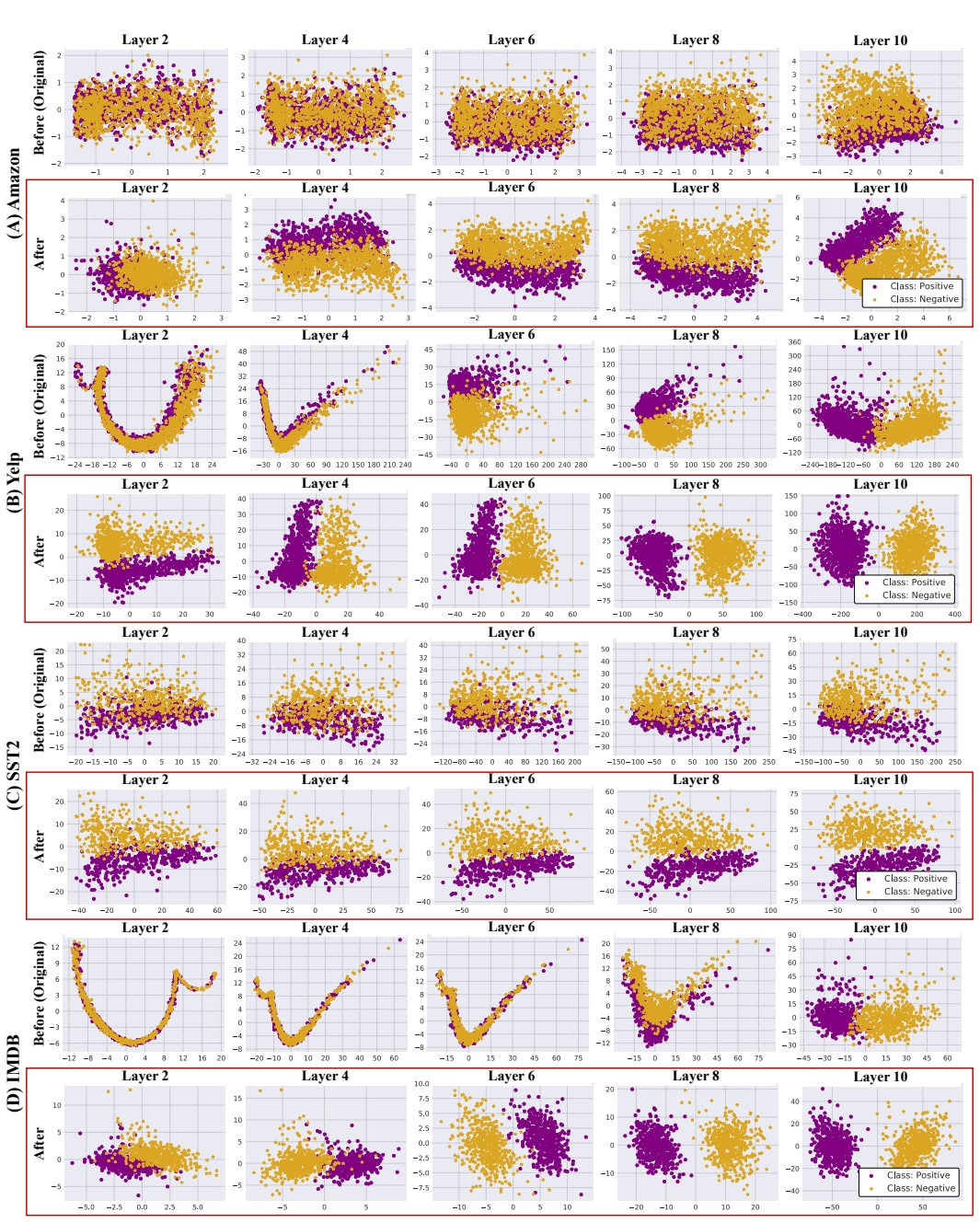

Figure 13: Visual representation of activations across five different layers of the **GPT-2** model for four different datasets: (A) Amazon, (B) Yelp, (C) SST2, and (D) IMDB. Odd-numbered rows show activations before applying Contrast-CAT's multiple contrasting, and even-numbered rows (highlighted in a red box) show activations after applying Contrast-CAT's multiple contrasting. The colors represent classes: positive (yellow) and negative (purple). Principal Component Analysis is used to reduce the dimensionality of activations to two dimensions for visualization. The separation between positive (yellow) and negative (purple) classes becomes more distinct after applying Contrast-CAT's multiple contrasting.

