# OpenReview forum: "Improving Transformer Interpretability with Activation Contrast-Based Attribution"
_ICLR.cc/2025/Conference — Submitted to ICLR 2025_

### Official Review · Reviewer_pm3n · 2024-10-19

**Soundness:** 2
**Presentation:** 2
**Contribution:** 2
**Rating:** 3
**Confidence:** 4

**Summary:**

This paper proposes a counterfactual explanation method that utilizes the model's activation and reference activation for better token attribution. Specifically, they leverage the conclusion from [1] that the model's prediction confidence for an input token at layer $l$ can be decomposed as the gradient times the subscription of its original activation and a contrastive activation from another token. The author then generalizes this lemma to obtain the attribution for each token by further multiplying the above value with the corresponding attention value. The proposed method achieves promising results in counterfactual explanation, which means it can highlight the most important tokens in a sentence, and removing them will let the model's confidence drop significantly. They use AOPC and LOdds to reflect the ratio of the number of important tokens removed and the confidence difference after removing those tokens, which are common metrics in XAI paper.

Refs:

[1] Sangkyun Lee and Sungmin Han. Libra-cam: An activation-based attribution based on the linear approximation of deep neural nets and threshold calibration. In IJCAI, pp. 3185–3191, 2022

**Strengths:**

- This paper addresses the important problem of explaining how a transformer works by following the line of works in the attribution-based method.
- The method is tested not only on one transformer model but across a variety of them, including BERTbase, DistilBERT, RoBERTa, and GPT-2, showing its robustness and generalizability across different architectures and datasets. This increases the broader applicability and utility of the approach for a wide range of NLP tasks.

**Weaknesses:**

- The method looks similar to [1] mentioned in the paper with minimal modification. The only thing the author did was multiply the attention value to the conclusion from [1] without a reasonable explanation.
- AOPC and LOdds results can be affected by many factors, including **(1) how to skip the special tokens like [SEP] and [PAD]** (the author can either not skip any special tokens during the backpropagation but remove them when calculating the normalized attribution scores or skip part of the special tokens during the backpropagation or skip all special tokens during backpropagation can lead to different results) and **(2) how to "remove" the important tokens** (replacing the token into [PAD] instead of removing it from the sentence can lead to different results). The author did not clarify these important points in their paper.
- The author only focuses on the classification tasks, which is not exciting. The author should at least discover more about text generation tasks.

Refs:

[1] Sangkyun Lee and Sungmin Han. Libra-cam: An activation-based attribution based on the linear approximation of deep neural nets and threshold calibration. In IJCAI, pp. 3185–3191, 2022

**Questions:**

- For the contribution (1), what do the authors mean by "class irrelevant" features?

---

### Official Review · Reviewer_7CWH · 2024-10-19

**Soundness:** 2
**Presentation:** 2
**Contribution:** 2
**Rating:** 6
**Confidence:** 3

**Summary:**

This paper introduces an activation-based attribution method, Contrast-CAT, to attribute the important tokens in text classification task. They compare their method Contrast-CAT with attention-based, LRP-based, and activation-based methods on BERT, where Contrast-CAT outperforms all the other methods.

The writing and organization of the paper is good.  And the proposed experiments can prove that the proposed method Contrast-CAT achieves better accuracy than the previous activation-based attribution method AttCAT.

**Strengths:**

1.	The writing and organization of the paper is good.
2.	The proposed method Contrast-CAT achieves better accuracy than the previous activation-based attribution method AttCAT.

**Weaknesses:**

1. My main concern is the speed of this method. The gradient-based methods are slow because they require many backward computations. If the method requires too many computations, it is very hard to be applied in large language models.
2. This work lacks the comparison with causal-based methods. To identify the important tokens, an easier and faster method is to replace each token with [MASK] and calculate the probability decrease of the predicted token. And more complicated causal-based methods are commonly used in LLMs, such as [1,2].

[1] Locating and Editing Factual Associations in GPT, 2022

[2] Towards Best Practices of Activation Patching in Language Models: Metrics and Methods, 2023

**Questions:**

1. Can you provide specific runtime comparisons between Contrast-CAT and other methods in your experiments? Can you provide the runtime of Contrast-CAT for larger models, such as Llama-7B?
2. Can you explain why you chose not to include causal-based methods in your comparisons? Can you provide the results of causal-based methods?
3. Attributing at token-level is not enough for interpretability. Is it possible to identify the important inner parameters (e.g. attention heads/FFN neurons) using Contrast-CAT?
4. Text classification is an easy task. How Contrast-CAT might perform on tasks like factual knowledge attribution?

---

### Official Review · Reviewer_3jLr · 2024-10-27

**Soundness:** 3
**Presentation:** 3
**Contribution:** 2
**Rating:** 5
**Confidence:** 3

**Summary:**

This paper presents Contrast-CAT, a method enhancing Transformer interpretability in NLP tasks. Contrast-CAT introduces an activation contrasting framework that filters out class-irrelevant features by subtracting reference activations from target activations at each Transformer layer, focusing on class-relevant information. This approach yields precise token-level attribution maps, with experiments showing that Contrast-CAT surpasses current methods in attribution quality and interpretability across diverse datasets and models.

**Strengths:**

● The paper is exceptionally clear, with well-organized explanations and effective visual aids that clarify complex methods and results.

● This paper introduces an original approach with activation contrast, effectively filtering out class-irrelevant features and enhancing interpretability in transformer models.

**Weaknesses:**

● The authors conduct experiments primarily on text classification datasets, which are relatively simple and similar in nature. It remains unclear whether this method is effective for more complex tasks, such as MMLU, MATH, or HumanEval. Exploring these could provide a more comprehensive evaluation of the approach.

● The applicability of this method to decoder-based models is not fully addressed. While the experiments focus on encoder-based transformers (e.g., BERT), with limited GPT-2 results included in the appendix, it would be valuable to see a more thorough evaluation on large language models (LLMs) that have become influential in the research community, such as LLaMA, Mistral, and Qwen.

● If this method is extended to LLMs, it would be helpful for the authors to clarify any potential adjustments or considerations required. Additional details on how this approach might be adapted or optimized for LLMs would enhance the paper’s impact and applicability.

If these issues are addressed, I will consider raising my score.

**Questions:**

No further issues.

---

### Official Review · Reviewer_CkGx · 2024-11-05

**Soundness:** 3
**Presentation:** 2
**Contribution:** 3
**Rating:** 6
**Confidence:** 3

**Summary:**

The authors propose a new method for identifying which tokens are most influential to a model's prediction. This method, which they call Contrast-CAT, focuses on estimating the per-token influence on the correct class. The method works by trying to remove features from model activations which are supposed to be irrelevant to the class of interest. In their experiments, they demonstrate that this method works well from quantitatively and qualitatively.

**Strengths:**

originality: to my knowledge, this is original work.
quality:
* across several standard text classification datasets, the method appears to work well.
* I was also impressed by the highlighted qualitative examples which show intuitively better attributions compared to existing methods.
* The comparison to existing methods is also quite thorough. Including ablations on using multiple layers and the "Same" vs "Random" vs "Contrast" experiments were also convincing on the effectiveness of using contrastive reference activations.
* the evaluation methods seem sound and grounded in methods from prior work.
clarity: the overall flow of the paper is well-laid out.
significance: attributing language model behavior to individual inputs is an important cornerstone in interpretability. By providing a new improved method for this goal, the contribution is significant.

**Weaknesses:**

Improvements to the clarity of the method are important for the reader's understanding. The ease of understanding is currently a bit lacking from my read; I list specifics in the Questions section below.

Regarding significance, the method and experiments are limited to classification problems using models from the BERT family and GPT2. Since currently the focus in LLMs is on autoregressive models this may be of limited use for interpreting more popular and cutting-edge models in generation use cases, e.g. Llama. If feasible, it would be nice to see how effective this method is for larger models like Llama.

**Questions:**

* How do you identify the activations $R^{\ell}_i$? How efficient is this process? Can you give an example of a token sequence $r$ which satifies $f_c(r) < \gamma$?
* The notation is a bit unclear. For example, $A^{\ell, h}$ appears to refer to attention outputs whereas $A^{\ell}$ appears to refer to an activation map. This makes it difficult to understand exactly how the method works.

---

### Meta-Review · Area_Chair_D4wn · 2024-12-23

**Metareview:**

This paper proposes Contrast-CAT, an activation-based attribution method for enhancing the interpretability of Transformer models in NLP tasks, particularly text classification. The method filters out class-irrelevant features by subtracting reference activations from target activations at each Transformer layer. The authors conduct experiments on multiple datasets and models, comparing Contrast-CAT with several existing methods and demonstrating its superiority in attribution quality and interoperability.  However, the concerns of this paper come from several aspects: 1) the authors conduct experiments primarily on text classification datasets, which may limit the understanding of the method's effectiveness for more complex tasks; 2) applicability of the method to decoder-based models is not fully addressed, and it would be valuable to see a more thorough evaluation on large language models like LLaMA, Mistral, and Qwen. 3) the method looks similar to a previous work with minimal modification.
After comprehensive discussions, many concerns are fixed but the novelty of this paper is still not that clear.  " The concept of "class irrelevant feature" is not the first time proposed in this paper; it cannot be treated as the main contribution. Therefore, the authors' main contribution should be adding attention to previous works. However, the motivation for adding attention is unclear."

**Additional Comments On Reviewer Discussion:**

Revewiers raised concerns from several aspects. 1) Limited Task Scope: Reviewers 3jLr and pm3n noted that the authors conduct experiments primarily on text classification datasets, which may limit the understanding of the method's effectiveness for more complex tasks. 2) Applicability to Decoder-Based Models: Reviewer 3jLr pointed out that the applicability of the method to decoder-based models is not fully addressed, and the authors conducted additional experiments using Llama2 and reported the results, demonstrating the applicability and effectiveness of Contrast-CAT on larger, autoregressive models. 3) Method Similarity: Reviewer pm3n raised concerns that the method looks similar to a previous work with minimal modification. The authors explained that while there may be some structural similarities, Contrast-CAT is fundamentally different in terms of target neural architectures, the types of information utilized, and how that information is processed. After heavy discussion, the last concern still is not fixed.

---

### Decision · Program_Chairs · 2025-01-22

Reject